# A pilot study of lymphodepletion intensity for peripheral blood mononuclear cell-derived neoantigen-specific CD8 + T cell therapy in patients with advanced solid tumors

Dandan Li [1,2,3,9], Chao Chen[4,5,9], Jingjing Li[1,2,3,9], Jianhui Yue [4,9], Ya Ding[1,2,3,9], Hailun Wang [4,9], Zhaoduan Liang[4], Le Zhang [4,6], Si Qiu[4], Geng Liu[4], Yan Gao[4], Ying Huang[4], Dongli Li[4], Rong Zhang[7], Wei Liu[1,2,3], Xizhi Wen[1,2,3], Bo Li[4], Xiaoshi Zhang [1,2,3,10] ✉, Xi Zhang [4,10] ✉ & Rui-Hua Xu [2,3,8,10] ✉

Currently, the optimal lymphodepletion intensity for peripheral blood mononuclear cell-derived neoantigen-specific CD8 + T cell (Neo-T) therapy has yet to be determined. We report a single-arm, open-label and non-randomized phase 1 study (NCT02959905) of Neo-T therapy with lymphodepletion at various dose intensity in patients with locally advanced or metastatic solid tumors that are refractory to standard therapies. The primary end point is safety and the secondary end points are disease control rate (DCR), progression-free survival (PFS), overall survival (OS). Results show that the treatment is well tolerated with lymphopenia being the most common adverse event in the highest-intensity lymphodepletion groups. Neo-T infusion-related adverse events are only grade 1–2 in the no lymphodepletion group. The median PFS is 7.1 months (95% CI:3.7-9.8), the median OS is 16.8 months (95% CI: 11.9-31.7), and the DCR is 66.7% (6/9) among all groups. Three patients achieve partial response, two of them are in the no lymphodepletion group. In the group without lymphodepletion pretreatment, one patient refractory to prior anti-PD1 therapy shows partial response to Neo-T therapy. Neoantigen specific TCRs are examined in two patients and show delayed expansion after lymphodepletion treatment. In summary, Neo-T therapy without lymphodepletion could be a safe and promising regimen for advanced solid tumors.

Adoptive T cell therapy (ACT) is becoming a powerful treatment option for cancer patients. These include adoptive transfer of chimeric antigen receptor (CAR) T cells, T cell receptor (TCR) T cells, and tumor-infiltrating lymphocytes (TILs). Many clinical studies have shown that adoptive transfer of neoantigen-specific tumor-infiltrating lymphocytes (TILs) could mediate durable regression of several types of solid tumors[1–4]. However, collecting TILs requires invasive sampling of tumor tissues, and is often limited by the relatively small size of resected tumor tissues, and some tumor sites are also not easily accessible[5]. Recently, several studies reported that T cell clones

recognizing neoantigens were found in the peripheral blood of cancer patients[6,7]. And T cells in peripheral blood lymphocytes (PBLs) could recognize the same P53 neoantigen as TILs did[8]. Therefore, PBL could be a reliable and noninvasive source of T cells for developing neoantigen-targeting ACT therapy to treat cancer patients.

Non-myeloablative lymphodepletion is frequently used prior to ACT to suppress the host immune system to decrease immunogenicity, increase the availability of IL2, IL7, IL15, etc., reduce regulatory T cells, and increase the persistence of infused T cells. Fludarabine and cyclophosphamide are the most used drugs for lymphodepletion chemotherapy. However, lymphodepletion can also lead to neurotoxicity, neutropenia, anemia, and a greater risk of infection. Currently, the dosing intensity of lymphodepletion varied significantly across different centers and clinical trials and appeared to be T cell product specific. No standardized protocol exists for lymphodepletion[9,10].

Here, we develop a method to generate personalized Neo-Ts (neoantigen-specific CD8 + T cells) from patients' peripheral blood mononuclear cells. We build a pipeline including neoantigen prediction by bioinformatic analysis and large-scale neoantigen-specific expansion of T cells in 25 days. To assess the clinical efficacy of Neo-T treatment and determine optimal lymphodepletion intensity for Neo-T therapy, we perform a first-in-human study of Neo-T therapy with dose escalation of fludarabine and cyclophosphamide. This study is designed as an open-label phase I clinical trial (NCT02959905) comprising patients with locally advanced or metastatic solid tumors refractory to standard therapies.

Here, we show that during Neo-T therapy, the different lymphodepletion dosages are well tolerated from no lymphodepletion to high-dose treatment. Neo-T therapy shows promising clinical efficacy in patients with advanced solid tumors with an ORR at 33.3% (3/9) and DCR of 66.7% (6/9). More patients in the no lymphodepletion group show PR during Neo-T therapy than in the higher intensity groups. Neoantigen-specific TCRs are examined in two responding patients and show delayed expansion after lymphodepletion treatment. Together, Neo-T therapy without lymphodepletion could be a safe and promising treatment strategy for advanced solid tumors.

## Results
### Patient characteristics
Eleven patients (median age 51 years; range 29-66 years) with locally advanced or metastatic cancer were enrolled at the Sun Yat-sen University Cancer Center between Feb.10, 2017, and Jun.19, 2019, including 9 patients with melanoma, 1 with colorectal cancer, and 1 with intrahepatic cholangiocarcinoma. All patients had been previously treated with at least one standard first-line therapy (Table 1). Patients were assigned to three treatment arms according to the time of enrollment. First three patients started with no LD chemo, the next three patients with low LD chemo, and the following three enrolled treated with high LD chemo (Fig. 1A, Fig. S1A). Next, all patients received Neo-T infusions at a frequency of 1 dose per month, and CT scan or MRI were performed every two months (Fig. 1A, B). Two patients in the no lymphodepletion group withdrew early during Neo-T treatment without CT evaluation. One patient had lung metastasis and plural effusion with grade II anemia before enrollment, their symptoms improved and felt less tired after the 1st Neo-T infusion, but their anemia progressively got worse and eventually had to withdraw after the 2nd Neo-T infusion. The other patient received 3 infusions of Neo-T, but withdrew due to a later discovered enrollment violation (Fig. 1A). Both patients were included in the adverse event (AE) analysis but excluded from the treatment efficacy analysis.

### Neo-T Generation and Infusion
Among the 11 patients enrolled, the median nonsynonymous tumor mutation burden (TMB) by whole-exome sequencing (WES) was 12.72

**Table 1 | Main clinical characteristics and clinical outcome of patients with advanced cancer treated with neo-T therapy**

| Patient No. | HLA-A | Cancer type | Stage at diagnosis | Sites of tumor | Prior therapies* | ECOG PS | Neo-T infusion (times) | Response (duration months) | Lymphodepletion (dose) |
|---|---|---|---|---|---|---|---|---|---|
| C001 | 0201;1102 | Melanoma | IV | Backside, lymph nodes, shoulder | Surgery,>2 lines of CT | 1 | 2 | NA/ withdrawn | N |
| C018 | 0101;1101 | ICC | IV | Liver, lymph nodes | Surgery, > 2 lines of CT | 1 | 3 | NA/ withdrawn | N |
| C003 | 1101;1102 | Melanoma | IV | Hip, lymph nodes, penis | Surgery,>2 lines of CT | 1 | 2 | PD | N |
| C004 | 1101;1102 | Melanoma | IV | Backside, bone, lung, kidney | Surgery,>2 lines of CT | 0 | 8 | PR (15) | N |
| C020 | 0201;3303 | Melanoma | IV | lymph nodes, adrenal gland | Surgery, >2 lines of CT and PD-1 | 1 | 15 | PR(27) | N |
| C005 | 0201;0203 | Melanoma | IV | Foot, bone, lymph nodes | Surgery, > 2 lines of CT | 1 | 8 | PR(8) | Low |
| C008 | 1101;0207 | Melanoma | IV | Vulva, lymph nodes, lung | Surgery, > 2 lines of CT | 0 | 3 | PD | Low |
| C012 | 1101;3303 | Colorectal cancer | IV | Colon, uterus, lymph nodes, hilar | Surgery, RT, 2 lines of CT | 1 | 6 | SD (4) | Low |
| C013 | 0206;1101 | Melanoma | IV | Foot, lymph nodes | Surgery, 2 lines of CT | 1 | 6 | SD (4.37) | High |
| C015 | 0201;1101 | Melanoma | IV | Thumb, lymph nodes | Surgery, 2 lines of CT | 1 | 6 | SD (4) | High |
| C017 | 1101;1101 | Melanoma | IV | Shoulder, lymph nodes | Surgery, 2 lines of CT | 1 | 2 | PD | High |

ICC intrahepatic cholangiocarcinoma, ECOG Eastern cooperative oncology group performance status, SD stable disease, PR partial response, PD progressive disease, CT chemotherapy, RT radiotherapy, N not pretreated, Low low dose level CTX and FDR; High high dose level CTX and FDR; NA not available due to withdraw from Neo-T therapy; N not pretreated.

A

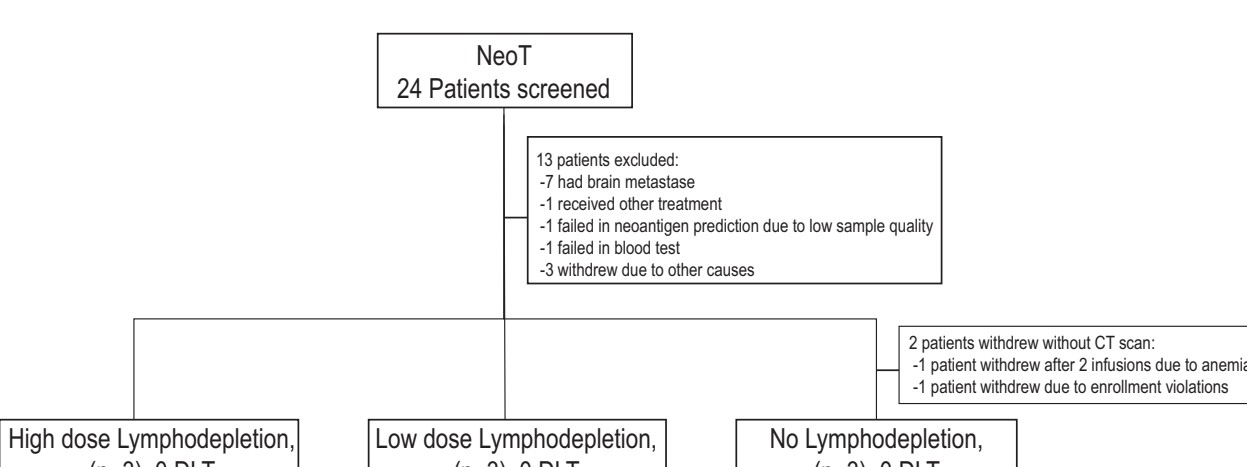

B

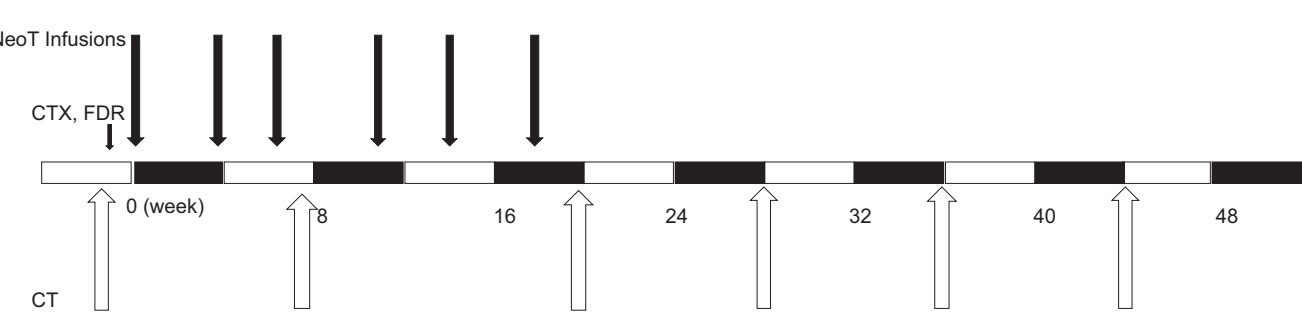

**Fig. 1 | Clinical trial diagrams of Neo-T therapy. A** Clinical trial flowchart. 24 patients were enrolled in this trial, and 11 patients received Neo-T therapy with different dose intensities of lymphodepletion. High dose group: both 500 mg/ m2 cyclophosphamide (CTX) and 25 mg/m2 fludarabine (FDR) for 2 days; Low dose group: 500 mg/m2 CTX for 1 day and 25 mg/m2 FDR for 2 days; and no lympho-depletion group. **B** Clinical trial scheme. Schematic depicting the phase I clinical trial. Nonmyeloablative lymphodepletion regimen was administered on days −5 and −4. Next, Neo-T cells were administered on Day 0. And 2 courses of Neo-T treatment were given to each patient, with 1 infusion/month and 3 infusions/ course. CT (Computed Tomography) scan was conducted every two months after the first cell infusion.

mutations per megabase of DNA (range 3.04–236.76) (Fig. 2A). And the number of potential neoantigens (IC50 < 500) were predicted to be between 22 and 1876 (Fig. S1B) using pipelines described in the "Methods" section. After HLA-A binding validation using T2 assay, 5 to 20 neoantigens were selected for PBMC derived CD8 + T cell activation and expansion (Table S1, Fig. S1C). Eventually, Neo-Ts were success-fully generated ex vivo from all 11 patients, and neoantigen-specific T cells in Neo-Ts were between 0.5% and 45% based on tetramer staining (Fig. 2A, Table S2). The median number of Neo-T cell infused was 1.57e8/dose (range 4.78e7 to 9.0e8). The cell viability and per-centage of CD8 + T cells were more than 90% in all the Neo-T products (Table S1). All Neo-T products were generated in our current good manufacturing practices (cGMP) -grade manufacturing facilities, and mycoplasma, bacteria, fungus were tested negative (Table S1). All patients received between 2-15 doses of Neo-Ts during the treatment (Table 1).

## Safety
The intensity of lymphodepletion we used was moderate even in our high intensity group at dosing levels of cyclophosphamide (500 mg x 2 days) and fludarabine (25 mg x 2 days). And lymphodepletion

itself was well tolerated. During Neo-T treatment, patients received a median of 6 doses of Neo-T administrations (range, 2 to 15), no acute adverse effects occurred. All treatment-related adverse events were listed in Table 2. There were six common adverse events that occurred in more than 20% of the patients, including anemia, lymphopenia, proteinuria, pruritus, rash, and sinus tachy-cardia (Table 2). Other adverse effects possibly resulting from Neo-T therapy, such as hypoalbuminemia, neutropenia et al., were grade 1 adverse events and disappeared in one week after cell infusion. Three patients had grade 2 adverse effects. Patient C005 had grade 2 sinus tachycardia after the last cell infusion. Patient C013 had grade 2 skin itches during administration and disappeared after 12 days. Patient C015 had grade 2 immune-associated pneumonia and the symptom disappeared after 8 days. Grade 3 or 4 toxicities were also observed during the treatment but were not related to Neo-T infusion. Patient C001 and patient C003 in the cohort with-out lymphodepletion developed grade 3 anemia attributable to prior chemotherapies. Grade 3 or 4 toxicities occurred more fre-quently in the low-intensity and high-intensity lymphodepletion groups, with lymphopenia being the most common adverse event (100%) (Table 2).

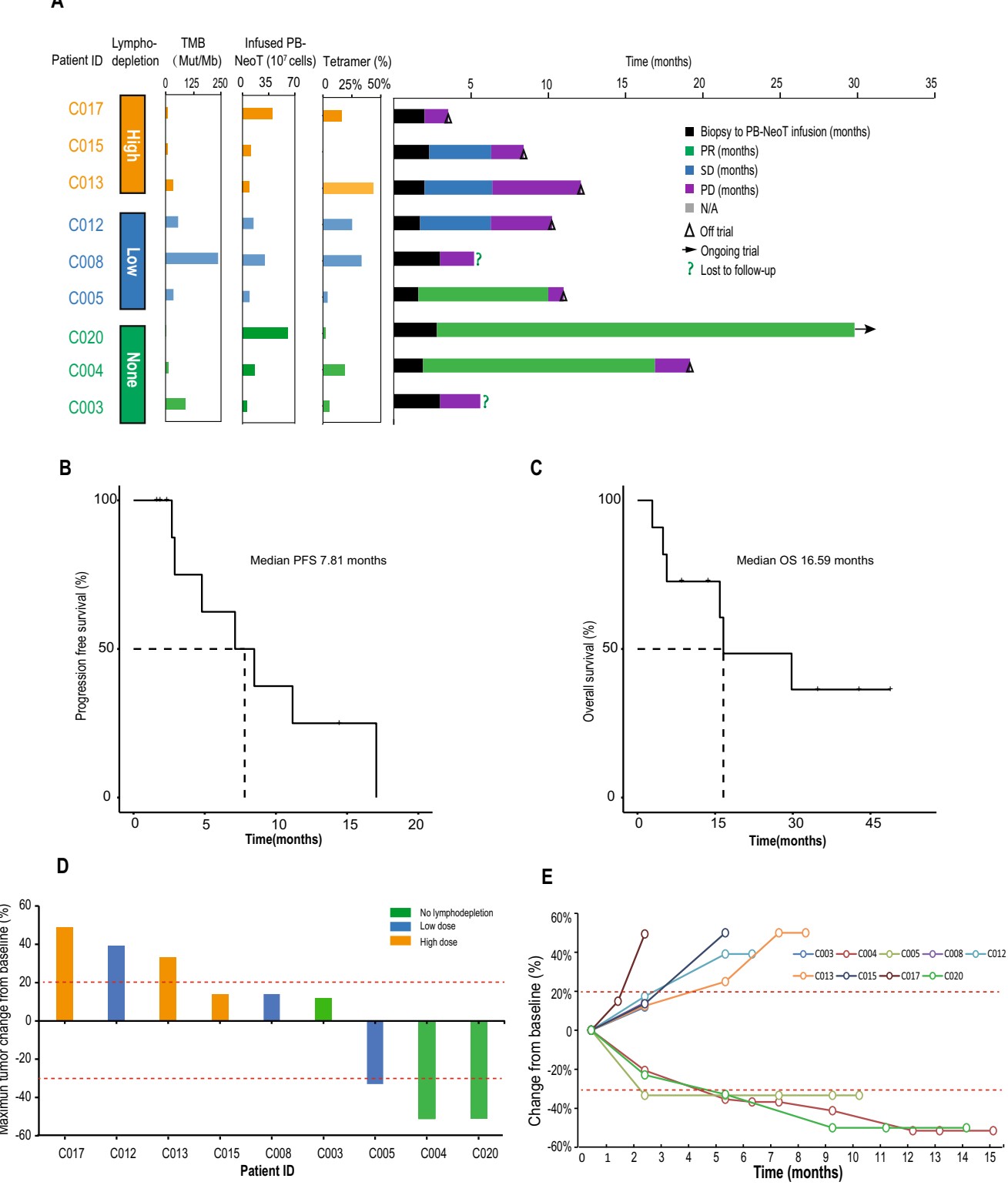

**Fig. 2 | Neo-T treatment and patient survival. A** Characteristics of Neo-T products, patient features and treatment outcomes in each patient. **B** Progression-free survival (PFS), and **C** Overall survival (OS) of all treated patients. **D** Waterfall plot showing the best overall change in sum of diameter of target lesions. **E** Percent change in sum of tumor diameters from baseline over time. CR, complete response; PR, partial response; SD, stable disease; TMB, tumor mutation burden; mut/MB, mutations per megabase; Tetramer%, percentage of neoantigen-specific T cells based on Tetramer staining. Source data are provided as a Source Data file.

## Clinical response

Patients enrollment and treatment scheme were shown in Fig. 1A, B. A total of eleven patients received Neo-T infusions, two patients (C001 and C018) withdrew early from the trial before CT evaluation because of anemia or other none treatment related causes (Fig.1A). Among the remaining nine patients, three patients had stable disease (SD) lasting a median of 4 months. Three patients had partial response. One of the PR patients - C005 was in the low-intensity

**Table 2 | Treatment-related adverse events**

| Adverse event/No. (%) | Lymphodepletion (dose) | | | | | |
| --- | --- | --- | --- | --- | --- | --- |
| | None (n = 5) | | Low (n = 3) | | High (n = 3) | |
| | Any grade | Grade ≥ 3 | Any grade | Grade ≥ 3 | Any grade | Grade ≥ 3 |
| Anemia | 2 (40.0) | 2 (40.0) | 3 (100.0) | 1 (33.3) | 3 (100.0) | 0 |
| lymphopenia | 0 | 0 | 3 (100.0) | 3 (100.0) | 3 (100.0) | 3 (100.0) |
| Leukopenia | 1 (20.0) | 0 | 2 (66.7) | 0 | 3 (100.0) | 0 |
| Neutropenia | 1 (20.0) | 0 | 3 (100.0) | 0 | 2 (66.7) | 0 |
| leukocytosis | 1 (20.0) | 0 | 0 | 0 | 0 | 0 |
| Neutrophilia | 1 (20.0) | 0 | 0 | 0 | 0 | 0 |
| Thrombocytopenia | 0 | 0 | 0 | 0 | 1 (33.3) | 0 |
| Hyperuricemia | 0 | 0 | 0 | 0 | 1 (33.3) | 0 |
| Pruritus | 1 (20.0) | 0 | 3 (100.0) | 0 | 1 (33.3) | 0 |
| Rash | 1 (20.0) | 0 | 2 (66.7) | 0 | 0 | 0 |
| Herpes zoster | 0 | 0 | 1 (33.3) | 0 | 1 (33.3) | 0 |
| Skin infection | 0 | 0 | 1 (33.3) | 1 (33.3) | 0 | 0 |
| Stomatitis | 0 | 0 | 1 (33.3) | 0 | 0 | 0 |
| Fever | 1 (20.0) | 0 | 0 | 0 | 0 | 0 |
| Acute bronchitis | 0 | 0 | 0 | 0 | 1 (33.3) | 0 |
| Cough | 2 (40.0) | 0 | 1 (33.3) | 0 | 1 (33.3) | 0 |
| Immune associated pneumonia | 1 (20.0) | 0 | 0 | 0 | 1 (33.3) | 0 |
| Sinus tachycardia | 0 | 0 | 2 (66.7) | 0 | 2 (66.7) | 0 |
| Gastrointestinal reaction | 1 (20.0) | 0 | 1 (33.3) | 0 | 2 (66.7) | 0 |
| Impaired liver function | 2 (40.0) | 0 | 1 (33.3) | 0 | 1 (33.3) | 0 |
| Creatinine increased | 1 (20.0) | 0 | 0 | 0 | 1 (33.3) | 0 |
| CPK increased | 2 (40.0) | 0 | 0 | 0 | 0 | 0 |
| Cystatin C increased | 1 (20.0) | 0 | 1 (33.3) | 0 | 2 (66.7) | 0 |
| Proteinuria | 2 (40.0) | 0 | 2 (66.7) | 0 | 2 (66.7) | 0 |
| Hypoalbuminemia | 2 (40.0) | 0 | 3 (100.0) | 0 | 3 (100.0) | 0 |
| Total bilirubin increased | 0 | 0 | 0 | 0 | 2 (66.7) | 1 (33.3) |
| Direct bilirubin increased | 0 | 0 | 0 | 0 | 1 (33.3) | 1 (33.3) |
| urobilinogen increased | 2 (40.0) | 0 | 0 | 0 | 1 (33.3) | 0 |
| Increased urine sediment mucus count | 1 (20.0) | 0 | 1 (33.3) | 0 | 2 (66.7) | 0 |
| Urine occult blood | 1 (20.0) | 0 | 3 (100.0) | 0 | 1 (33.3) | 0 |
| Fecal occult blood | 2 (40.0) | 0 | 1 (33.3) | 1 (33.3) | 1 (33.3) | 0 |

lymphodepletion group, they had a metastatic lesion in inguinal lymph node. Neo-T infusions led to a quick reduction in tumor size. The patient remained in PR for 9 months until a new lesion appeared in the ileum (Figs. 2A and 3). The other two PR patients were in the no lymphodepletion pretreatment group. Patient C004 achieved partial response (PR) 7 months after the first infusion. And two of their four target lesions (lesions 2 and 4) showed complete regression at 13 months, while the other two lesions were in slow regression. This patient's pain from right leg and right hip disappeared after six infusions of Neo-T cells. However, a new adrenal lesion developed 15 months later (Figs. 2A and 3). Patient C020 had been given anti-PD-1 antibody (Toripalimab), but a lesion in the left adrenal gland developed during ICB (immune checkpoint blockade therapy). After joining our trial, they responded to Neo-T treatment quickly, and their tumor lesions shrank after 6 infusions. They were later given 9 more infusions of Neo-Ts and remained in PR for over 27 months till the time we start preparing the manuscript (Figs. 2A and 3). It is worth noting that among this small number of patients, more patients in the no lymphodepletion and the low-intensity lympho-depletion groups showed PR response than in the high intensity group (Fig. 2A, D, E, Table 1).

Three patients did not respond to Neo-T treatment. Patient C003 withdrew from treatment after the 2nd infusion of Neo-T due to progressive anemia. Two other patients (C008, C017) were taken off trial after 2nd or 3rd infusion of Neo-T due to disease progression. Among 9 patients evaluated, the disease control rate (DCR) was 66.7% (6/9), and the objective response rate (ORR) was 33.3% (3/9) (Fig. 2A). By the time we prepared the manuscript, the median progression free survival (PFS) was 7.1 months (95% CI:3.7–9.8), and median overall survival (OS) was 16.8 months (95% CI: 11.9–31.7) (Fig. 2B, C).

**Neoantigen specificities of infused Neo-T products**

To confirm that the Neo-Ts produced from each patient could recognize predicted neoantigens, we examined Neo-Ts with tetramer staining. Results showed that about 0.4–43.9% of Neo-Ts could recognize their corresponding neoantigens in each patient (Fig. 2A, Table S2). To our surprise, infusion of more neoantigen specific T cells does not necessarily lead to better clinical outcomes (Fig. 2A).

We further studied the kinetics of neoantigen-specific T cells in two patients (004, 005) with confirmed partial response after Neo-T treatment. First, to identify neoantigen-specific Neo-T cell clones, three predicted neoantigens / mutant peptides in each patient that showed positive response in ELISPOT assay were selected (Fig. 4A), and their corresponding MHC tetramers were generated and used to enrich neoantigens specific Neo-T cells. Next, we performed single cell TCR sequencing on these enriched Neo-Ts using 10X genomics

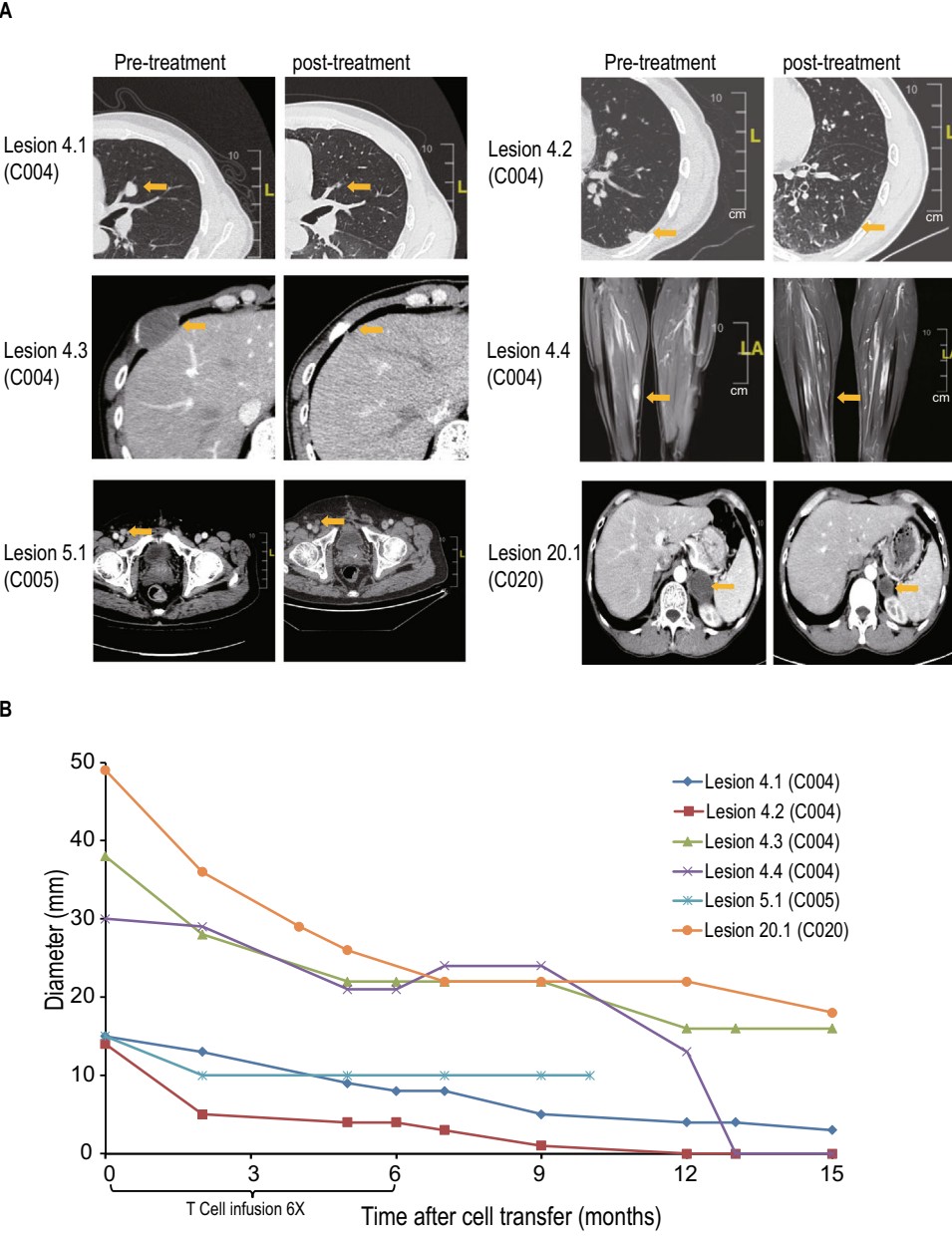

**Fig. 3 | Examples of Neo-T mediated tumor regression. A** Computed tomographic (CT) or magnetic resonance imaging (MRI) scans of patients C004, C005, and C020. Arrows highlight target lesions before and after treatment. **B** Change in diameters of target lesions from baseline over time. Source data are provided as a Source Data file.

platform, and identified 341 or 1739 TCR clones reactive to mutant peptides No. 4 or No. 5 from Patient C004, and 702, 717, or 462 TCR clones that were reactive to mutant peptides No. 1, No. 13, or No. 19 from Patient C005 (Table S3). There were a few dominant TCR clones existed in each individual group which suggested the successful expansion of several major TCR clones (Fig. 4C). The gene sequence of the top 3 TCRs from each target were cloned into a viral vector and transduced CD8+ T cells to generate neoantigen specific TCR-Ts for validation (Fig. 4B, C). Eventually, three TCR-T clones from each patient (004 or 005) showed strong neoantigen-specific IFNγ secretion and T2 cell killing and were selected as representative neoantigen specific TCRs for following in vivo analysis (Fig. S2, Fig. 4D).

To track the neoantigen-specific T cell in the peripheral blood, PBMCs were collected from patients 004 and 005 at different time points before and after Neo-T infusions and single-cell TCR sequencing were performed. Results showed that the frequency of those

representative neoantigen-specific TCRT clones increased significantly ( ~ 2–100 folds) after Neo-T infusions. Most TCRT clones contracted after 2-4 months since the start of the treatment, clone C004-5-1 and C005-13-1 persisted after the last dose of Neo-T infusion. From the results of C004, we also see two neoantigen-specific T cell clones (#4-1, #4-2) expanded shortly after the 1st NeoT infusion, while the third clone (#5-1) only start to expand after the 3rd Neo-T infusion (C004 in Fig. 4E). Therefore, the difference in expansion of the neoantigen-specific T cell also existed in the patient received no LD chemo, which suggested that there were potentially many factors including LD chemo could affect the dynamics of neoantigen-specific T cell clonal expansion after infusion. It will be difficult to find the correlation between T cell clonal expansion and the intensity of lymphodepletion regimens using our small sample size. But we do find a delayed expansion of all three TCR clones in patients with lymphodepletion pretreatment (C005 in Fig. 4E). These suggest lymphodepletion could potentially affect the

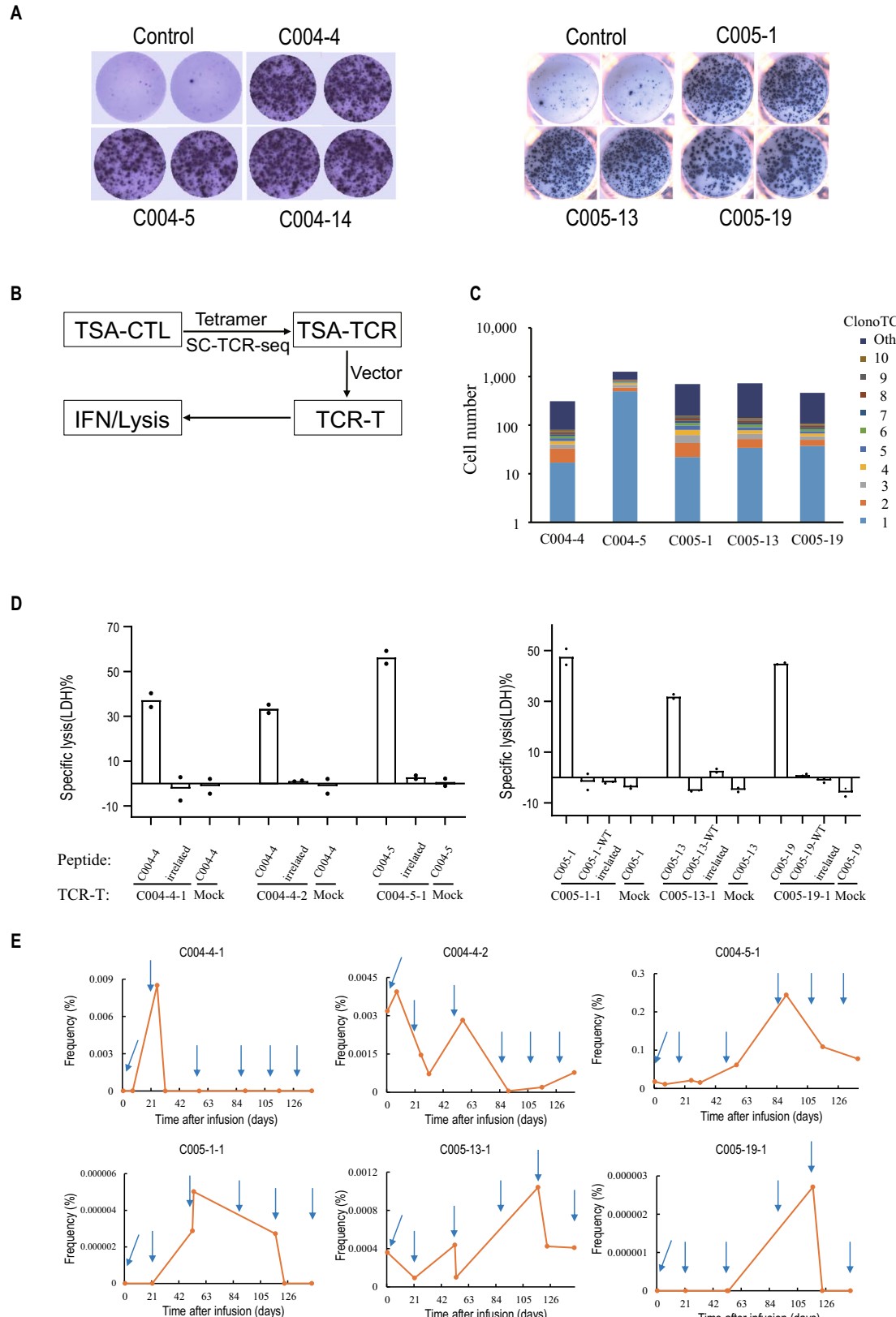

**Fig. 4 | Neoantigen-specific T cells in patient C004 and C005. A** Neo-T cells from patient C004 and C005 were stimulated with three selected neoantigens and IFN-γ secreting cells were detected by ELISPOT assay. **B** Schematic diagram of TSA-specific TCR cloning and functional validation. TSA, tumor specific antigen. **C** Top 10 TCR clones in neoantigen-specific T cells. **D** TCR-T cells mediated specific lysis of T2 cells loaded with corresponding mutant peptides. Irrelated: irrelated peptide; WT: wild type peptide; Mock: GFP-T cells. Data shown here are the mean value of two replicates. **E** Frequency of several representative neoantigen-specific TCR clones in peripheral blood of patient C004 and C005 during Neo-T treatment. Arrow, Neo-T infusion. Source data are provided as a Source Data file.

expansion of infused Neo-T cells, but more data are needed to validate these findings. (Fig. 4E).

Lymphodepletion is known to have a long-term impact on lymphocyte levels. In our study, it took 1–3 months for the patients' lymphocyte count to recover to baseline (Fig. S4), which coincident with the expansion of neoantigen-specific TCR clones in patient C005 (Fig. 4E). Therefore, one dose of lymphodepletion could potentially affect the expansion of multiple doses of Neo-T cells, but more data are needed to validate these findings.

## Discussion

Non-myeloablative lymphodepletion has several beneficial properties during ACT, including reducing the number of immunosuppressive cells such as Tregs and bone marrow-derived suppressive cells, increasing availability of interleukin IL2, IL7, and IL15 and enhancing the persistence of adoptively transferred T cells. However, lymphodepletion suppresses host immune system and can lead to severe neutropenia, lymphopenia, anemia, and a greater risk of infection. How to achieve a balance between efficacy and safety from lymphodepletion is of great interest in ACT field. In a recent study, Nissani and colleagues analyzed data from three phase 2 trials of adoptive cell therapy including melanoma patients receiving TILs therapy[11]. What they found was that a lower dose of lymphodepletion regimen ($2 \times 30$ mg/kg cyclophosphamide and $5 \times 25$ mg/m2 fludarabine) during preconditioning were comparably effective in achieving deep lymphopenia but less toxic than a higher dose ($2 \times 60$ mg/kg cyclophosphamide and $5 \times 25$ mg/m2 fludarabine). However, no patients in the lower dose group achieved objective response to TIL therapy[11].

So far, the majority of data about effective lymphodepletion regimens came from Car-T and TILs therapies, and appeared to be T cell product specific. In the present study, we set out to determine the optimal lymphodepletion intensity for Neo-T therapy. We chose a commonly used lymphodepleting regimens of cyclophosphamide (500 mg/m2) and fludarabine (25 mg/m2), but at a reduced dosage – both 500 mg/m$^2$ cyclophosphamide and 25 mg/m$^2$ fludarabine for 2 days in high dose group; and 500 mg/m$^2$ cyclophosphamide for 1 day and 25 mg/m$^2$ fludarabine for 2 days in low dose group; and no lymphodepletion group. We demonstrated that Neo-T treatment with lymphodepletion at various intensity were well tolerated. Because lymphodepletion can also effectively decrease host lymphocyte count and reduce the immunogenicity of injected T cells, there is a greater risk of potentially severe AEs associated with Neo-T infusion in the no lymphodepletion group. But in our study, no grade 3 and above AEs were associated with Neo-T infusion in the no lymphodepletion groups. High-intensity, low intensity and no lymphodepletion treatments had similar safety profiles. It is worth noting that in our study, Neo-T infusions without lymphodepleting pretreatment also induced tumor regression in patient C004 and C020. Furthermore, despite repeated administration of Neo-T cells without further lymphodepletion treatment, toxicity was not observed among all patients. Meanwhile, multiple infusions promoted expansion of several neoantigen-specific TCR clones, suggesting beneficial boosting effects with multiple treatments of Neo-T (Fig. 4E). TCRseq data also showed persistence of several neoantigen-specific TCR clones in responding patients during the six-months treatment, suggesting a potentially long-lasting Neo-T immune responses in cancer patients. But more TCR sequencing and immunogenic data (ELISPOT) from patient beyond 6 months will be needed to confirm this, which we did not collected in the current phase I study. Overall, Neo-T therapy at current treatment regimens and dosages appeared to be safe in heavily pretreated patients.

In this study, we showed that Neo-T could be generated successfully from PBMC in all patients and recognized several predicted neoantigens. Although infused at relative small dosages between 4.78e7 to 9.0e8, Neo-T treatment showed a promising 33% ORR comparable to TIL/TCRT therapies which had infused T cell numbers often in the range of 1e10 to 1e11. Unexpectedly, the group of patients with no lymphodepletion chemotherapy had the best clinical responses (Fig. 2A). One plausible explanation is that LD chemo creates an unfavorable microenvironment in the body for Neo-T proliferation and activation (Fig. 4E, Fig. S4). Therefore, without LD chemo, a lower number of Neo-T could achieve clinical responses comparable to other adoptive T cell transfer studies with LD chemo and with more T cells administered. While our findings showed that LD chemo reduces lymphocyte counts, inhibits and delay the expansion of neoantigen-specific TCR clones, and suggest that LD may reduce the efficacy of Neo-T treatment, it is possible that other factors could have contribute to the outcomes. For example, differences in patient characteristics, the tumor microenvironment, the immunogenicity of neoantigens, and the amount of neoantigen-specific Neo-T cells infused could all have influenced the results. Limited by the size of the current study, we were unable to investigate the effects of all these variables and to generalize our findings to a larger population. Therefore, future investigations with larger sample sizes are necessary to better understand the impact of lymphodepletion on Neo-T treatment responses.

We noticed there were several patients who received a similar number of T cells with a high ratio of neoantigen specificity, but did not respond to Neo-T therapy (Fig. 2A). One possible reason was an immune suppressive tumor environment, so a combination therapy with ICB treatment could potentially improve the efficacy of Neo-T treatment. Also, a higher infusion number of T cells combined with IL2 treatment may also help to increase treatment efficiency.

Finally, the Neo-T therapy presented in this study has several advantages: (1) T cells are derived from PBMC and are easier to obtain than TIL[12]; (2) Neoantigens are only expressed in tumors and have less off-target toxicity or adverse effects than Tumor-associated antigens (TAA) (such as the *MAGEA*)[13]. (3) We only need 100-300 M T cells for infusion, which is much less than the number administered in other adoptive T cell therapies reported to treat solid tumors[3,14–16]. Currently, with the continuous improvement in the predictive accuracy of immunogenicity of neoantigens, Neo-T therapy could be a promising approach in cancer immunotherapy.

In summary, this study demonstrated that adoptive transfer of Neo-Ts derived from peripheral blood lymphocytes is safe in various lymphodepletion regimens, and is able to mediate durable tumor regression without lymphodepletion chemotherapy.

## Methods
### Study design
This is an open-label phase I clinical trial that used a 3 + 3 dose escalation strategy to determine the optimal lymphodepletion intensity for Neo-T therapy in patients with locally advanced or metastatic solid tumors that are refractory to standard therapy (NCT02959905). The clinical trial protocol was approved by Institutional Ethics Committee at the Sun Yat-sen University Cancer Center, and was conducted in accordance with the principles of the Declaration of Helsinki. All patients provided written informed consent before enrollment. The primary objective of this study was to determine safety and tolerability of various intensity lymphodepletion, and the secondary objective was to assess clinical responses of Neo-T therapy, including disease control rate (DCR), progression-free survival (PFS), overall survival (OS) and duration of response (DOR). However, at the end of the trial, only three patients' data could be used for the DOR analysis (C004, C005, C020 in Fig. 2A), so we did not include DOR analysis in the manuscript. The exploratory objective was to study the kinetics of infused neoantigen-specific T cells to understand and improve Neo-T therapy.

For pretreatment conditioning, a nonmyeloablative regimen was administered consisting of cyclophosphamide (CTX) and fludarabine (FDR) at different doses: (1) no lymphodepletion group: with no CTX

and FDR pretreatment; (2) low-dose group: with CTX 500 mg/m$^2$ for one day intravenously (i.v.; day −5) and FDR 25 mg/m$^2$ per day for 2 days (i.v.; days −5 and −4); (3) high-dose group: CTX 500 mg/m$^2$ for 2 days (i.v.; days −5 and −4) and FDR 25 mg/m2 per day for 2 days (i.v.; days −5 and −4).

Because all patients enrolled in our trial had surgery and more than 2 lines of chemotherapies, we were concerned about whether the patients could tolerate multiple rounds of lymphodepleting chemotherapy (LD chemo), so we decided to give only one dose of LD chemo to patients before the 1st Neo-T infusion, and also use a mild LD chemo regimen. Next, Neo-T was administered on Day 0. And 2 courses of Neo-T treatment were given to each patient according to the clinical trial protocol, with 1 infusion/month and 3 infusions/course. After completion of the two courses of treatment, patients could choose to receive additional cycles of consolidation treatments of Neo-T.

In the safety assessment, adverse events were categorized and graded according to Common Terminology Criteria for Adverse Events (CTCAE) Version 5.0. CT (Computed Tomography) or MRI (Magnetic Resonance Imaging) was conducted every two months after the first cell infusion. Clinical efficacy was evaluated according to Response Evaluation Criteria in Solid Tumors (RECIST) 1.1. Because most PD patients missed the CT scan required by iRECIST, we did not perform iRECIST assessment. Unfortunately, due to funding limitations, Part2-the expansion study of this Phase I trial was not/will not be performed. We hope to continue the study in the future once we secure more funding.

### Patients

Eligible patients should satisfy the following conditions: (1) 18 to 70 years of age with a pathologically confirmed diagnosis of metastatic or locally advanced cancer; (2) progressed after receiving at least one standard first-line treatment; (3) have an Eastern Cooperative Oncology Group (ECOG) performance status of no greater than 1; (4) HIV antibody negative, Treponema pallidum negative, Hepatitis C virus antibody negative, and HBV DNA negative; (5) HLA type is HLA-A11:01+ or HLA-A02:01 + . Eleven patients (median age 51 years; range 29-66 years) with locally advanced or metastatic cancer were enrolled at the Sun Yat-sen University Cancer Center between Feb.10, 2017, and Jun. 19, 2019. The patient characteristics are reported in Table 1. Patients received Neo-Ts as single-agent treatment throughout the study period.

### Sequencing and analysis

DNA and RNA were extracted from FFPE tumor tissues using RecoverAll™ Total Nucleic Acid Isolation Kit (Invitrogen, Cat#:AM1975), libraries construction, and sequencing (pair-end 150 bp) were conducted according to protocols on Illumina platform as previously described[17,18]. Low-quality reads were removed with SOAP nuke v1.5[19]. Whole-exome sequencing data were aligned to reference genome (hg19) using DRAGEN pipeline. Furthermore, indel-realignment and recalibration were performed by the Genome Analysis Toolkit (GATK)[20,21]. MuTect[22] and Strelka[23] were used to identify SNVs and InDels, respectively. RNA sequencing reads were mapped to reference genome through STAR (version 2.5.3a)[24]. Gene expression (Transcripts Per Million, TPM) was calculated by using RSEM[25].

Single-cell VDJ and 5′ transcriptome libraries of CD8$^+$tetramer$^+$ cells were constructed following protocols, and sequenced on HiSeq X instruments (Illumina) platform (pair-end 150 bp). Raw data processing was performed using the Cell Ranger version 2.1 pipeline (10x Genomics). The 'vdj' and 'count' of Cell Ranger were used to analyze the expression of gene and identify TCRs, respectively. TCRs were further compared with the PIRD database (https://db.cngb.org/pird/tbadb/) to remove the pathogen-specific TCRs, and the obtained mutant epitope-specific TCRs.

### Neoantigen prediction

Mutant peptides were generated from nonsynonymous SNVs or InDels with VAF (mutated allele frequency in tumor) > 0.01. Mutant peptides which have same sequences with peptides from NCBI Reference Sequence Database were filtered out. Affinity of each mutant peptides and corresponding HLA alleles from same patient was predicted by NetMHC3.0[26], NetMHCpan4.0[27], PickPocket[28], PSSMHCpan[29], and SMM[30]. For each peptide, there are five affinity values (IC50). Its final IC50 will be the smallest value out of the five IC50s if there are three or more values with IC50s < 500, otherwise its IC50 will be 50,000. For SNVs, the affinities of corresponding wild type peptides were also predicted. We filtered out a mutant peptide if: IC50 > 500 nM, or IC50$_{mutant}$ > IC50$_{widetype}$, or TPM of its source gene <1. Mutant peptides were ranked according to their IC50.

### Dendritic cells maturation

CD8+ cells were obtained using CD8 microbeads (Miltenyi Biotec) from PBMCs, and cryopreserved. The remaining cells (PBLs) were resuspended in DC medium (CellGenix) to achieve a concentration of about 2e6 cells per ml. They were incubated for 2 h at 37 °C to allow adherence to flask/dish. Nonadherent cells were washed out. Clinical grade human granulocyte-macrophage colony stimulating factor (GM-CSF, 800IU/ml; PrimeGene) and interleukin 4 (IL-4, 1000IU/ml; PrimeGene) were added. Cells were incubated for 2 days. CD40L (200 ng/ml, Bio-Techne) and IFN-γ (400IU/ml, PrimeGene) were added on day 3. Mutant peptides were loaded on day 4. Mature dendritic cells (DCs) were collected on day 5.

### Neo-T generation

Matured DCs and pre-thawed CD8$^+$ cells were cocultured at a 1:4 ~ 10 ratio in AIM-V medium with 2% autologous serum and IL21 (30 ng/ml, CellGenix) in flask (T75 or T175) for two or three days. IL2 (40 ~ 50IU/ml, CellGenix), IL-7 (5 ~ 10 ng/ml, PrimeGene), and IL15 (1 ~ 2 ng/ml, PrimeGene) were added into the medium. Then cells were incubated for 8–10 days. These T cells were re-stimulated again, and incubated for the next 8–10 days. T cells were collected and transferred into infusion bags for treatment.

### Enzyme-Linked ImmunoSPOT (ELISPOT) Assay

IFNγ ELISPOT kits, including pre-coated plates and alkaline phosphatase (ALP), were bought from Mabtech. The assay was conducted following their protocols. ELISPOT assay strip plate was washed with PBS and blocked with RPMI 1640 containing 10% FBS. T cells were cocultured with peptide only or T2 cells pre-pulsed with or without peptides (10 µg/ml) in the above pre-treated ELISPOT plate for 24 hours. The plate was rinsed with PBS, then added with alkaline phosphatase (ALP) labeled anti-human IFNγ mAb (7-B6-1-ALP, 1:200; Mabtech) for 2 hours. After rinsing, 5-Bromo-4-chloro-3-indolyl phosphate/Nitro blue tetrazolium (BCIP/NBT, Mabtech) was used to develop the immune-spot. Spots were imaged and counted by an ELISPOT Reader (BioReader 4000, BIOSYS).

### Flow cytometry assay

Cells were collected, washed, and resuspended in FACS buffer, and stained with fluorescent dye conjugated antibodies for 15 minutes at 4 °C. Peptide-MHC tetramers were generated through ultaviolet-irradiation-mediated peptide exchange method[31]. APC-Cy™7 Mouse Anti-Human CD3 (BD Biosciences, 557832, 2.5 µl/5 × 10$^5$ cells), Alexa Fluor® 700 Mouse Anti-Human CD8 (BD Biosciences, 557945, 2.5 µl/5 × 10$^5$ cells), and Flex-T™ HLA-A*02:01 Monomer UVX (BioLegend, Cat#280004, 0.05 µg/1 × 10$^6$ cells) were used. After washing twice in FACS buffer, cells were analyzed using a FACSAria II (BD Biosciences) with live cell gating based on 4′,6-diamidino-2-phenylindole (DAPI) exclusion, and CD8$^+$ pMHC-tetramer$^+$ cells were sorted. The data were analyzed using FlowJo software (Tree Star).

## TCR transduced T cells

To identify neoantigen specific TCR pairs in each group, we built TCR cassette with *TRAV*, *TRBV*, and mouse conserved regions of TCR, then constructed lentiviral vectors to transduced CD8 + T cells. CD8⁺ T cells were enriched using CD8 MicroBeads (Miltenyi Biotec) from healthy donors, maintained in RPMI-1640 medium (Gibco) supplemented with 10% FBS (Gibco) and IL2 (100IU/ml, Peprotech), and stimulated by CD3/CD28 microbeads (Gibco) for 24 h. Activated CD8⁺ cells were transduced by lentiviral particles containing TCR gene in the presence of protamine (10 μg/ml, sigma), and then incubated in a 37 °C 5% $CO_2$ incubator for 7 days. The expression rate of TCR on T cells was determined by staining with APC anti-mouse TCRβ chain Antibody (BioLegend).

## Cytotoxicity assay

The neoantigen specificity of TCR-T cells were determined by cytotoxic activity and IFNγ production. TCR-T cells were co-cultured with T2 cells pre-pulsed with or without peptides (10 μg/ml) in 96-well round-bottom plate for 24 h at the E:T rate 5:1. The supernatant was collected and lactate dehydrogenase (LDH) was detected by the CytoTox 96® Non-Radioactive Cytotoxicity Assay kit (Promega). In brief, culture supernatant and the CytoTox 96® reagent was mixed as equal aliquot in 96-well flat-bottom plate, and incubated in room temperature for 30 min. The reaction was stopped by stop solution, and the quantity of formazan product, which was positive correlation with the cytotoxicity, was detected through the absorbance at 490 nm within 1 hour after adding the stop solution. The percentage lysis was calculated according to the manual.

## High throughput sequencing and analysis of TCR repertoire

The third complementary determining region (CDR3) of TCRs were amplified by multiplex PCR and sequenced. Briefly, gDNA(1200 ng) for each sample were amplified using QIAGEN Multiplex PCR Kit (QIAGEN) with 32 forward primers annealed to the FR3 region and 13 reverse primers annealed to the junction (J) region of TCR. The target amplified product (100–200 bp) was purified by electrophoresis on 2% agarose gel and then were sequenced with single end 150 bp reads on BGIseq500 platform.

Sequencing data were analyzed by IMonitor[32]. A brief summary of the pipeline was as follows: (1) The low-quality reads and reads containing adapters were filtered to get clean reads. (2) The cleaned reads were merged, and the reads that could not be merged were discarded. (3) The merged reads were aligned to their respective V, D, J germline sequences (IMGT, http://www.imgt.org/) using BLAST. (4) Realignment of the correctly mapped reads was performed to select the best V/D/J alignment. (5) Further statistical analysis were performed by R software.

## Statistical analysis

In ELISPOT analysis, a positive response was the number of spots (more than 10 spots) detected at least two-fold of the negative control[33]. In TCR-T cytotoxic activity test, the statistical analysis and graphical presentations were computed by GraphPad Prism software (GraphPad Software Inc.). The data were expressed as the mean ± SEM ($n = 3$). Unpaired student's t-test was used to determine the statistical significance, in which 2-sided $P < 0.05$ was considered significant. Overall survival (OS) was defined as the time from the date of first cell reinfusion to death from any cause or the last known follow-up date. Progression-free survival (PFS) was defined as the time from the first infusion to the presence of disease progression or death from any cause. The 52-month OS and PFS were evaluated with the Kaplan–Meier method.

## Reporting summary

Further information on research design is available in the Nature Portfolio Reporting Summary linked to this article.

## Data availability

The sequencing data can be accessed through GSA under the accession code HRA004715. Sequencing data are available under restricted access. Access can be obtained by completing the application form via GSA-Human System and/or by contacting the corresponding authors. The clinical Study Protocol Synopsis is available as Supplementary Note in the Supplementary Information file. Other individual de-identified participant data will be shared upon request from the corresponding authors. The remaining data are available within the Article, Supplementary Information or Source Data file. Source data are provided with this paper.

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

## Acknowledgements

We sincerely thank the support provided by China National GeneBank and BGI-GenoImmune. This work was supported by the National Natural Science Foundation of China (NO. 81772910). This work was also supported in part by the Basic and Applied Basic Research Foundation of Guangdong Province under Grant No. 2022A1515111138.

## Author contributions

Conception and design: B.L., XS.Zh., X.Zh., RH. X. Collection and assembly of data: XS. Zh, DD. L., JJ.L., Y.G., XZh. W., Y.D., HL.W. Data analysis and interpretation: C.C., HL.W., ZhD.L., JH.Y., L.Zh., W.L., Y.H., S.Q., G.L., DL.L., R.Z., and X.Zh. Manuscript writing: HL.W. and all authors. Final approval of manuscript: All authors.

## Competing interests

The authors declare no competing interests.

## Additional information

[1]Biotherapy Center, Sun Yat-sen University Cancer Center, 510060 Guangzhou, China. [2]State Key Laboratory of Oncology in South China, 510060 Guangzhou, China. [3]Collaborative Innovation Center for Cancer Medicine, 510060 Guangzhou, China. [4]BGI-Shenzhen, Shenzhen 518083, China. [5]Department of Thoracic Surgery, Peking University Shenzhen Hospital, Shenzhen Peking University-The Hong Kong University of Science and Technology Medical Center, Shenzhen 518035, China. [6]BGI Education Center, University of Chinese Academy of Sciences, Shenzhen, China. [7]Department of Radiology, Sun Yat-sen University Cancer Center, Guangzhou, China. [8]Department of Medical Oncology, Sun Yat-sen University Cancer Center, 510060 Guangzhou, China. [9]These authors contributed equally: Dandan Li, Chao Chen, Jingjing Li, Jianhui Yue, Ya Ding, Hailun Wang. [10]These authors jointly supervised this work: Xiaoshi Zhang, Xi Zhang, Rui-Hua Xu. ✉e-mail: zhangxsh@sysucc.org.cn; zhangxi1@cngb.org; xurh@sysucc.org.cn

