## [Peer Review File · Nature Communications]

A Pilot Study of Lymphodepletion Intensity for Peripheral Blood Mononuclear Cell-Derived Neoantigen-Specific CD8+ T cell Therapy in Patients with Advanced Solid TumorsREVIEWER COMMENTS

Reviewer #1 (Remarks to the Author): with expertise in adoptive cell therapy, melanoma, immunotherapy

This manuscript describes the results of a small phase I study using an autologous peripheral blood derived non-gene modified neoantigen-specific cell product in combination with different lymphodepletion regimens: no LD chemo; low intensity LD chemo; high intensity LD chemo. The primary endpoint was the safety and tolerability of the LD chemo schedules and the secondary endpoint ORR and PFS of the neoAg T cell treatment. The authors showed no real safety concerns between the groups, except typical adverse events coming from LD chemo. Partial responses were observed in the no LD chemo and low intensity chemo cohorts.

The question at stake is important as the cell therapy field is struggling with whether or not and how much LD chemo to give.

The study is remarkable in many ways:

1.) the LD chemotherapy provided was not very intense: days -5, -4 either CTX 500mg/m² for 1 or 2 days and days -5, -4 fludarabine 25 mg/m². The LD chemo was followed by cell infusion on day 0 and every month thereafter but as far as I understood all subsequent doses of T cells were given without prior LD chemo.

Is this correct??

If so, and because the first read-out was after 8 weeks (following cell infusion) than all patients also had T cell infusions with NO LD chemo.

The median number of doses were 6.

2.) The cell dose was: 1.5×10^8 NeoAg T cells per dose. These cells were generated from PBMC, and if I am not mistaken the production did not take longer than 4 weeks (two rounds of stimulation lasting each 10-14 days), which is impressive.

3.) in the 9 evaluable patients, 3 had PD as best response, and the other had clinical benefit (3x SD lasting 4 months and 3x PR up to 9 months). 2 PRs were in the No LD chemo cohort, which is highly interesting

Comments:

I find it difficult to understand the trial design. Why was the LD chemo only given once? The majority of patients had melanoma and had not been treated with chemotherapy. Therefore this could have been repeated. Why was chosen for a very mild LD chemo regimen. Not sure, the difference in intensity between the 2 LD chemo cohorts was significant or clinically relevant. In both cases it was low intensity LD chemo.

There was quite a difference in expansion of the neoAg T cells between the groups after the first dose, and some of the expansions occurred late. It will be hard to correlate this to the LD chemo or no chemo regimens.

When were the responses observed? Early on, or also late? This is important as except for dose 1, all other cell infusions were not preceded by LD chemo. How should the results than be interpreted? ORR and SD can occur without or with very mild LD chemo.

Although this is a very important finding, I am not sure this was meant by the authors.

In analysing the T cell specificities, between 0.4 and 43% of neoAg T cells (within the infusion product?) recognised the neoAg. If this is the case, how many neoAg T cells were actually infused? Was the sum in all patients 1.5×10^8 ? Please explain?

As the neoantigens were predicted by algorithms, it is very feasible that these may not be presented by the tumor cells (not processed). Although it is nice to show that T cells can recognise peptide-loaded T2 cells, this is not proof that the T cells can kill endogenous tumors. Were the T cells tested

against tumor digests or cell lines? Why not? This would strengthen the results.

The authors demonstrate clonal expansion, sometimes up to 100x. Also contraction after 2-4 months. An obvious question is whether this was correlated with response or also observed in SD patients (or even PD patients)?

What was the mechanism of escape following SD or PR? Was there loss of MHC or antigen (immune editing)?

Reviewer #2 (Remarks to the Author): with expertise in adoptive cell therapy, tumor reactive T cells

This manuscript presents the results of a clinical study aimed at evaluating responses of patients with metastatic cancer to adoptive transfer of tumor-reactive T cells generated from the peripheral blood by in vitro stimulation (IVS). There were several issues, however, with the results that were reported in this study.

1. There are several issues regarding the clinical parameters used to carry out the trial. It was not clear how patients were assigned to treatment arms and it is not clear if progression following the most recent treatments were documented.
2. The longest ongoing responder had received anti-PD1 therapy and had a single adrenal metastasis, but this was not biopsied and so, as pointed out in the text, the change in size could have resulted from adrenal hemorrhage. Another responder, patient C005, only had a relatively small lymph node lesion that demonstrated a small degree of shrinkage that was of questionable significance, and lesion 4.3 in patient 4 could also have resulted from reabsorption of fluid or a hemorrhage.
3. It appeared that the neoantigen responses were evaluated retrospectively but the data shown in Table S1 was not clearly explained. It is not clear how the Average number of Neo-Ts were calculated, and it is unclear why an average value was presented as this presumably should be a single number for an individual patient. The Average percentage of CD8+ T in Neo-Ts values were all close to 1, which is difficult to understand as there is no reason why all these values should be approximately 1, unless this represents the frequency. If this represents the frequency, that is also not credible as this degree of enrichment is very unlikely to results from 2 rounds of IVS.
4. The total number of administered T cells and in particular neoantigen-reactive T cells is extremely low and is unlikely to have any impact on disease burden, given the results of previous adoptive transfer studies in patients with solid cancers where on the order of $1e10$ or more T cells have generally been administered.

Reviewer #3 (Remarks to the Author): with expertise in biostatistics, clinical trial study design

This manuscript is a phase I trial of lymphodepletion intensity for Neo-T therapy in patients with advanced solid tumors. It used 3+3 design with 3 dose cohorts (no lymphodepletion, low-intensity and high-intensity lymphodepletion). The primary endpoint is safety and secondary endpoint is overall response rate.

1) What's the order of the does escalation? Start with "no" and increase to "low" and "high"?

No grade 3+ Neo-T infusion related AE was observed in 3 cohorts, 2 PR in "no", 1 PR in "low" and no PR in "high" was observed. It's interesting that "high" group has worst response.

2) Both "low" and "high" cohort have 100% grade 3+ lymphopenia, are their lymphocyte counts different? What about other cell counts?

3) The PFS and OS are not clearly defined in the methods. Line 238, "the 52-month overall survival rate and recurrence rate was evaluated with the Kaplan–Meier method". But there is no such rate in the results. Instead, the median PFS and OS are reported. What's median follow-up time?

4) line 238, "the $p < 0.05$ was considered significant" should be "2-sided $p < 0.05$ "

5) line 330, "no direct correlation was found between the frequency of neoantigen-specific T cells and clinical response in the current study". Is there a statistical test? Please report the p-value.

6) line 356, "The kinetics of TCRTs in patient 004 (no lymphodepletion group) and 005 (low-dose lymphodepletion group) were similar and did not seem to be affected by lymphodepletion intensity". Is there a statistical test? Please report the p-value.

7) line 367, "Nissani and colleges analyzed data from three phase 2 trials of", please add the citation.

8) Fig 4E c004-4-1 has different x-axis than other plots

The reviewers' comments and suggestions are very constructive and helpful. In this revision, according to reviewers' suggestions, we have supplemented new data with analysis and corrected several mistakes in our previous draft. The following are our point-to-point responses to the reviewers' comments.

Reviewer #1 (Remarks to the Author): with expertise in adoptive cell therapy, melanoma, immunotherapy

This manuscript describes the results of a small phase I study using an autologous peripheral blood derived non-gene modified neoantigen-specific cell product in combination with different lymphodepletion regimens: no LD chemo; low intensity LD chemo; high intensity LD chemo. The primary endpoint was the safety and tolerability of the LD chemo schedules and the secondary endpoint ORR and PFS of the neoAg T cell treatment. The authors showed no real safety concerns between the groups, except typical adverse events coming from LD chemo. Partial responses were observed in the no LD chemo and low intensity chemo cohorts.

The question at stake is important as the cell therapy field is struggling with whether or not and how much LD chemo to give.

The study is remarkable in many ways:

1.) the LD chemotherapy provided was not very intense: days -5, -4 either CTX 500mg/m² for 1 or 2 days and days -5, -4 fludarabine 25 mg/m². The LD chemo was followed by cell infusion on day 0 and every month thereafter but as far as I understood all subsequent doses of T cells were given without prior LD chemo.

Is this correct??

If so, and because the first read-out was after 8 weeks (following cell infusion) than all patients also had T cell infusions with NO LD chemo.

The median number of doses were 6.

Response: Yes, all subsequent doses of T cells were given without prior LD chemo in the low intensity and high intensity LD chemo groups.

2.) The cell dose was: 1.5 x 10⁸ NeoAg T cells per dose. These cells were generated from PBMC, and if I am not mistaken the production did not take longer than 4 weeks (two rounds of stimulation lasting each 10-14 days), which is impressive.

Response: Yes, we managed to reduce the number of production days. Currently, the production took 22-25 days.

3.) in the 9 evaluable patients, 3 had PD as best response, and the other had clinical benefit (3x SD lasting 4 months and 3x PR up to 9 months). 2 PRs were in the No LD chemo cohort, which is highly interesting

Comments:

I find it difficult to understand the trial design. Why was the LD chemo only given once? The majority of patients had melanoma and had not been treated with chemotherapy. Therefore this could have been repeated. Why was chosen for a very mild LD chemo regimen. Not sure, the difference in intensity between the 2 LD chemo cohorts was significant or clinically relevant. In both cases it was low intensity LD chemo.

Response: Thank the reviewer for the comment. As we know, LD chemo using cyclophosphamide (CTX) and fludarabine (FDR) is associated with significant side effects, such as bone marrow toxicity, fevers, neurotoxicity, pericarditis, hemorrhagic cystitis, etc. And all patients enrolled in our trial already had surgery and more than 2 lines of chemotherapies (Table 1). When we designed the trial, we were concerned about whether the patients could tolerate multiple rounds of LD chemo, so we decided to give only one dose of LD chemo to patients before the 1st Neo-T infusion. We added this information to the "Study Design" in the main text in this revision. (lines: 110-114)

Currently, the dosing intensity of LD chemo varied significantly in different adoptive cell therapy clinical trials, and appeared to be T cell product specific. In most trials, the dose of FDR was used at 25/30 mg/m² daily for 2-5 days, but the dose of CTX varied considerably in the intensity, ranging from 250-750mg/m² daily for 1-3 days (PMID: 30501490, PMID: 28291388, PMID: 27605551, Clinical Trial: NCT02869217, etc.). Based on our previous experience, CTX at 500mg/m² daily for 2 days and FDR 25mg/m² daily for 3 days could lead to serious adverse event (SAE), such as febrile neutropenia in some patients. Therefore, to ensure the safety of patients, we reduced the dose of FDR by one day, and used an LD chemo regimen with CTX 500 mg/m² and FDR 25 mg/m² per day for 2 days in the high-intensity cohort.

Our previous experience tells us that one-day CTX at 500mg/m² is also effective for lymphodepletion (can also be seen in Fig.S4). So, in the low-intensity cohort, we reduced the dose of CTX by 50% to 500mg/m² for 1 day, and maintained the same dose of FDR at 25 mg/m² per day for 2 days.

There was quite a difference in expansion of the neoAg T cells between the groups after the first dose, and some of the expansions occurred late. It will be hard to correlate this to the LD chemo or no chemo regimens.

Response: Thank the reviewer for the comment. With limited resources and samples, we only performed neoantigen-specific TCR identification and TCR sequencing in two patient samples (C004- no LD chemo, and C005- low LD chemo). From the results of C004, we can see two neoAg T cell clones (#4-1, #4-2) expanded shortly after the 1st NeoT infusion, while the third clone (#5-1) only start to expand after the 3rd Neo-T infusion (C004 in Fig.4E below). Therefore, the difference in expansion of the neoAg T cell also existed in the patient received no LD chemo, which suggested that there were potentially many factors including LD chemo could affect the dynamics of neoAg T cell clonal expansion after infusion. It will be difficult to find the correlation between T cell clonal expansion and the intensity of LD chemo regimens using our small sample size. But we do find a delayed expansion of all three neoAg TCR clones in LD chemo

patient (C005). These suggest LD chemo could potentially affect the expansion of infused Neo-T cells, but more data are needed to validate these findings. We added the discussion to the Result section in this revision. (lines: 373-384)

Figure 4E (in main text): Frequency of several representative neoantigen-specific TCR clones in peripheral blood of patient C004 and C005 during Neo-T treatment. Arrow, Neo-T infusion.

When were the responses observed? Early on, or also late? This is important as except for dose 1, all other cell infusions were not preceded by LD chemo. How should the results than be interpreted?

Response: Thank the reviewer for the comments. In our study, the first clinical evaluation was performed in the 8th week (after two doses of Neo-T) when patients already showed response to the treatment (Fig.2E in main text). LD chemo is known to have a long-term impact on lymphocyte level. In our study, it took 1-3 months for the patients' lymphocyte count to recover to baseline (Fig.S4, below), which coincident with the expansion of neoAg T cell clones in patient C005 (Fig.4E, above). Therefore, one dose of LD chemo could potentially affect the expansion of multiple doses of Neo-T cells, but more data are needed to validate these findings. We added this explanation to the Result section. (lines:385-390)

ORR and SD can occur without or with very mild LD chemo. Although this is a very important finding, I am not sure this was meant by the authors.

Response: Thank the reviewer for the comments. We changed the statement in the "Abstract" from: "In summary, Neo-T therapy with low intensity lymphodepletion or no lymphodepletion could be a safe and promising regimen for advanced solid tumors." to "In summary, Neo-T therapy without lymphodepletion pretreatment could be a safe and promising regimen for advanced solid tumors."

FigureS4 (in supplementary data): Total lymphocyte counts in different patients before and after lymphodepletion

In analysing the T cell specificities, between 0.4 and 43% of neoAg T cells (within the infusion product?) recognised the neoAg. If this is the case, how many neoAg T cells

were actually infused? Was the sum in all patients 1.5×10^8 ? Please explain?

Response: Thank the reviewer for the comment. Yes, these numbers are the percentage of T cells in the infusion products that recognized our predicted neoantigens. The sums were listed in the Table X1 below, and they are not the same across all these patients. However, there were potentially some T cells in the product that could recognize other tumor antigens than the ones we predicted. So, it will be difficult to know the exact number of tumor-specific T cells in our products.

Because patients had different types and quantities of neoantigens, and these neoantigens expressed and processed differently and had different immunogenicity. So, technically, it is difficult to produce the same number of neoAg-specific T cells across all patients.

Meanwhile, we also examined whether the number of infused neoAg-specific T cells correlated with the clinical response. One-way logistics regression was used to calculate the correlation and the results showed a P value of 0.387, suggesting no significant correlation in our current dosing regimen (Table X1).

Table X1. Number of neoAg specific-T cells in infusion products

Patient No.	Number of neoAg specific-Ts (* 10^8)	Clinical Response	P value
C003	0.85	PD	0.387
C004	3.93	PR	
C005	0.44	PR	
C008	6.76	PD	
C012	2.62	SD	
C013	6.94	SD	
C015	0.05	SD	
C017	5.32	PD	
C020	2.15	PR	

As the neoantigens were predicted by algorithms, it is very feasible that these may not be presented by the tumor cells (not processed). Although it is nice to show that T cells can recognise peptide-loaded T2 cells, this is not proof that the T cells can kill endogenous tumors. Were the T cells tested against tumor digests or cell lines? Why not? This would strengthen the results.

Response: Yes, we agree with the reviewer that the predicted neoantigens may not be presented by the tumor cells, and a direct test of T cell killing on tumor digests will strengthen the results. However, in our study, these patients had surgery a while ago (<2 years), and we could only use tumor FFPE tissue blocks for DNA/RNA extraction, sequencing and neoantigen prediction, no fresh tumor tissues were available for us to test. Also it's practically difficult to get enough fresh tumor sample for testing in every patients. As to the cell lines, they usually don't have the same neoantigens and HLA alleles as in patient samples, so it's difficult to find a cell line for testing.

The authors demonstrate clonal expansion, sometimes up to 100x. Also contraction after 2-4 months. An obvious question is whether this was correlated with response or also observed in SD patients (or even PD patients)?

Response: Thank the reviewer for the comment. The clonal expansion was observed in C004 and C005 patients with clinical response (Fig.2E, Table1, and Fig.4E). But unfortunately, we had limited resources and samples, and only performed neoantigen-specific TCR identification and TCR sequencing in these two patient samples. We don't have the clonal expansion data in other patients.

What was the mechanism of escape following SD or PR? Was there loss of MHC or antigen (immune editing)?

Response: This is definitely an interesting question. But we did not collect tumor tissues after immune escape in this trial, so we don't know the escaping mechanisms. We will include this study in the future trials.

Reviewer #2 (Remarks to the Author): with expertise in adoptive cell therapy, tumor reactive T cells

This manuscript presents the results of a clinical study aimed at evaluating responses of patients with metastatic cancer to adoptive transfer of tumor-reactive T cells generated from the peripheral blood by in vitro stimulation (IVS). There were several issues, however, with the results that were reported in this study.

1. There are several issues regarding the clinical parameters used to carry out the trial. It was not clear how patients were assigned to treatment arms and it is not clear if progression following the most recent treatments were documented.

Response: Thank the reviewer for the comments. Patients were assigned to three treatment arms according to the time of enrollment. First three enrolled patients started with no LD chemo, next three patients with low LD chemo and the following three treated with high LD chemo. We clarified this in the manuscript. (lines: 264-267)

Yes, the data of progression following the most recent treatments were shown in the Table.X2 below, and patients in the no LD chemo groups showed the longest progression free time.

LD Chemo	Patient ID	Date of Most Recent Cell Infusion	Date of Disease Progress	Days
None	C020	2019.08.21	N/A (PR, no progression)	N/A
	C003	2018.01.08	2018.02.07	30
	C004	2018.11.02	2019.05.14	183
Low	C005	2018.11.13	2018.11.20	7
	C008	2018.03.05	2018.03.07	2
	C012	2018.10.08	2018.11.25	48
High	C013	2018.12.04	2019.01.07	34
	C015	2018.11.08	2018.11.20	12
	C017	2018.11.05	2018.11.08	3

2. The longest ongoing responder had received anti-PD1 therapy and had a single adrenal metastasis, but this was not biopsied and so, as pointed out in the text, the change in size could have resulted from adrenal hemorrhage. Another responder, patient C005, only had a relatively small lymph node lesion that demonstrated a small degree of shrinkage that was of questionable significance, and lesion 4.3 in patient 4 could also have resulted from reabsorption of fluid or a hemorrhage.

Response: Thank the reviewer for the comment.

1). In general, adrenal hemorrhagic tumors are characterized by areas of high density or stratification (hematoma stratification) within the mass on CT scans, and their size can change rapidly (Fig. X1). In the longest ongoing responder (C020), the patient's left adrenal lesion gradually grew larger over six months before Neo-T treatment, and the density of the tumor was homogenous on CT scans (Figs. X1, X2). Therefore, the tumor does not conform to the imaging characteristics of hemorrhagic lesions. As to adrenal cysts, which also manifest as homogenous low attenuation masses on CT images, they tend to grow slowly and a large cyst like the one in C020 is unlikely to resolve spontaneously. Therefore, considering this patient had melanoma, the radiologist diagnosed the lesion as metastatic melanoma.

Meanwhile, other than the representative left adrenal lesion showed in the manuscript, C020 patient had bilateral adrenal lesions. The right adrenal lesion is smaller, and also shrunk after the Neo-T treatment, from 21mm to 13mm (Fig.X3), which further demonstrated the efficacy of Neo-T treatment.

2). The small lymph node lesion in patient C005 shrunk from 15mm to 10mm, which is a 33.3% reduction in size, and was considered as partial response.

3). Similar to our response in #1) above, in general, soft tissue hemorrhagic lesions are characterized by high-density areas or stratification of lesions on CT scans, and their size can rapidly increase or decrease. As to lesion 4.3 in patient C004, during baseline radiographic evaluation, the lesion showed slightly increased enhancement on the contrast-enhanced CT scan. Swollen soft tissue shadow was visible around this right 6th rib lesion, and the size of soft tissue shadow gradually reduced after Neo-T infusion. Furthermore, the density of the lesion was homogenous. Later, after the gradual reduction of soft tissue shadows, destruction of the 6th rib on the right side is clearly visible (Fig.X4), so lesion 4.3 does not conform to the reabsorption process of a hematoma, or a cyst.

FigureX1. a). Example of CT scan image of an adrenal hemorrhage (white arrow) from the literature. The hemorrhagic foci show heterogeneous intensity, with areas of high density or stratification visible within the mass (PMID: 27738706). b). CT scan of the left adrenal lesion (red arrow) in patient C020 before Neo-T infusion. The density of the tumor was homogenous.

FigureX2. CT scans of the left adrenal lesion at different time points before and after Neo-T treatment.

FigureX3. Contrast-enhanced CT scan showing the right adrenal lesion (arrow) before and after Neo-T treatment.

FigureX4. CT scans of the lesion 4.3 in patient C004 before and after Neo-T treatment.

3. It appeared that the neoantigen responses were evaluated retrospectively but the data shown in Table S1 was not clearly explained. It is not clear how the Average number of Neo-Ts were calculated, and it is unclear why an average value was presented as this presumably should be a single number for an individual patient. The Average percentage of CD8+ T in Neo-Ts values were all close to 1, which is difficult to understand as there is no reason why all these values should be approximately 1, unless this represents the frequency. If this represents the frequency, that is also not credible as this degree of enrichment is very unlikely to results from 2 rounds of IVS.

Response: Thank the reviewer for the comments. Sorry for the confusion. Table S1 characterized the properties of infusion Neo-T products. In this trial, each patient received 6 doses of Neo-T infusions. The production and filling process will generate slight variation in cell numbers between each bag, so we presented an average number in the table. As to the CD8+ cells, the values should be the ratio, not percentage of CD8+ T cells in the final product. We corrected it in the Table. Our process of Neo-T cell production required isolating CD8+ T cells from PBMC as the starting materials, so our final product usually contain 90%-100% of CD8+ T cells.

4. The total number of administered T cells and in particular neoantigen-reactive T cells is extremely low and is unlikely to have any impact on disease burden, given the results of previous adoptive transfer studies in patients with solid cancers where on the order of $1e10$ or more T cells have generally been administered.

Response: Thank the reviewer for the comments. We were also surprised to see that the low number of Neo-T cells ($\sim 5e8$) could achieve clinical responses in several patients. Furthermore, the group of patients with no lymphodepletion chemotherapy had the best clinical responses (Fig. 2A). One plausible explanation is that LD chemo creates an unfavorable microenvironment in the body for Neo-T proliferation and

activation (Fig.4E, Fig.S4). Therefore, without LD chemo, a lower number of Neo-T could achieve clinical responses comparable to other adoptive T cell transfer studies with LD chemo and with more T cells administered. Certainly, for the treatment of solid tumors, other factors like the immunogenicity of neoantigens, the tumor microenvironment and some unknown factors could all contribute to the clinical responses. Therefore, a larger scale study is needed to provide clear evidence. We added this to the Discussion section. (lines: 434-445).

Reviewer #3 (Remarks to the Author): with expertise in biostatistics, clinical trial study design

This manuscript is a phase I trial of lymphodepletion intensity for Neo-T therapy in patients with advanced solid tumors. It used 3+3 design with 3 dose cohorts (no lymphodepletion, low-intensity and high-intensity lymphodepletion). The primary endpoint is safety and secondary endpoint is overall response rate.

1) What's the order of the dose escalation? Start with "no" and increase to "low" and "high"?

No grade 3+ Neo-T infusion related AE was observed in 3 cohorts, 2 PR in "no", 1 PR in "low" and no PR in "high" was observed. It's interesting that "high" group has worst response.

Response: Yes, considering the toxicity related to lymphodepletion chemotherapy, we designed the trial with dose escalation that started with "no" and increased to "low" and "high". We clarified this in the manuscript. (Lines: 264-267)

Yes, we were also surprised to find that "high" group has worst response. LD chemo is known to have a long-term impact on lymphocyte level. In our study, it took 1-3 months for the patients' lymphocyte count to recover to baseline (Fig.S4), which coincident with the expansion of Neo-T cell clones in patient C005 (low LD chemo) (Fig.4E). Therefore, one dose of high LD chemo could potentially delay the expansion of multiple doses of Neo-T cells and reduce the treatment efficacy. But more data are needed to validate these findings.

2) Both "low" and "high" cohort have 100% grade 3+ lymphopenia, are their lymphocyte counts different? What about other cell counts?

Response: Thank the reviewer for the comments. The cell counts data were shown in the Table X3 below. There is no significant difference in lymphocytes, white blood cells, neutrophils, lymphocytes, red blood cells, monocytes, platelets, and hemoglobin between the low-intensity and high-intensity lymphodepletion groups (student's unpaired t-test).

Table X3. Lymphocyte and other cell counts after LD chemo

	Patient	White blood cells (10 ⁹ /L)	Neutrophil (10 ⁹ /L)	Lymphocyte (10 ⁹ /L)	Hemoglobin (g/L)	Red blood cell (10 ¹² /L)	Monocyte (10 ⁹ /L)	Platelet (10 ⁹ /L)
Low-intensity lymphodepletion	C005	2.58	2	0.3	135	4.65	0.2	184
	C008	4.24	3.6	0.3	124	4.18	0.1	323
	C012	3.44	2.8	0.1	108	4.05	0.2	212
High-intensity lymphodepletion	C013	2.75	2.1	0.3	134	4.56	0.2	149
	C015	3.07	2.76	0.11	109	3.42	0.11	195
	C017	5.17	4.8	0.13	107	3.4	0.19	225
P-value	/	0.80	0.67	0.52	0.35	0.14	1.00	0.35

3) The PFS and OS are not clearly defined in the methods. Line 238, “the 52-month overall survival rate and recurrence rate was evaluated with the Kaplan–Meier method”. But there is no such rate in the results. Instead, the median PFS and OS are reported. What’s median follow-up time?

Response: Thank the reviewer for the comment. 1). In our trial, overall survival was defined as time from the date of first cell reinfusion to death from any cause or the last known follow-up date. Progression-free survival was defined as the time from first infusion to the presence of disease progression or death from any cause. We added the definition to the Method section in this revision (lines:246-250). 2). Sorry for the mistake, it’s not “overall survival rate and recurrence rate”, it should be “overall survival and progression-free survival were evaluated with the Kaplan-Meier method”. We corrected it in the manuscript. 3). And the median follow-up time is 8.6m (95%CI 1.3-52).

4) line 238, “the p<0.05 was considered significant” should be “2-sided p<0.05”

Response: Thanks, we corrected it in the manuscript.

5) line 330, “no direct correlation was found between the frequency of neoantigen-specific T cells and clinical response in the current study”. Is there a statistical test? Please report the p-value.

Response: Thank the reviewer for the comment. One-way logistics regression was used to calculate the correlation between the amount of neoantigen-specific T cells infused and the clinical response, and the results showed a P value of 0.387, suggesting no significant correlation. However, the sample size was small, and we need more data before we can draw any conclusions. So in this revision, we changed the statement to “Infusion of more neoantigen specific T cells does not necessarily lead to better clinical outcomes” (lines:348-349)

6) line 356, “The kinetics of TCRTs in patient 004 (no lymphodepletion group) and 005

(low-dose lymphodepletion group) were similar and did not seem to be affected by lymphodepletion intensity”. Is there a statistical test? Please report the p-value.

Response: Thank the reviewer for the comment. Sorry, our previous statement is not accurate. We changed it to “From these data, we can see the difference in expansion of the neoAg T cell even existed in the patient received no LD chemo treatment (C004 in Main Text Fig.4E). Therefore, there were potentially many factors including LD chemo could affect the dynamics of neoAg T cell clonal expansion after infusion. It will be hard to find the correlation between this and the intensity of LD chemo regimens using our small sample size. But we do find a delayed expansion of all three neoAg TCR clones in LD chemo patient (C005, Main Text Fig.4E). These suggest LD chemo could potentially affect the expansion of infused Neo-T cells, but more data are needed to validate these findings.” We added this new state in this revision. (lines:373-384)

7) line 367, “Nissani and colleges analyzed data from three phase 2 trials of”, please add the citation.

Response: Thank the reviewer for the comment. We added the citation: “Nissani A, Lev-Ari S, Meirson T, Jacoby E, Asher N, Ben-Betzalel G, Itzhaki O, Shapira-Frommer R, Schachter J, Markel G, Besser MJ. Comparison of non-myeloablative lymphodepleting preconditioning regimens in patients undergoing adoptive T cell therapy. *J Immunother Cancer*. 2021 May;9(5):e001743.”

8) Fig 4E c004-4-1 has different x-axis than other plots

Response: Thank the reviewer for pointing it out. We removed the minor ticks on the x-axis in Fig4E c004-4-1.

REVIEWERS' COMMENTS

Reviewer #1 (Remarks to the Author):

I would like to thank the authors for their clear answers, which is very helpful

Reviewer #2 (Remarks to the Author):

The changes made in the manuscript addressed the majority of issues raised by the reviewers, but there are a few points that need to be further addressed.

1. It is not possible with a small study such as this to say that the LD had any effect on responses with the limited number of patients in this study.
2. The information presented in Table S1 needs further clarification. If this is the fraction of CD8+ T cells in the infused products then it should be noted as such and not a ratio. A ratio compares 2 different values and the legend just refers to a single variable, the number of CD8+ T cells.
3. It is not clear why the viability of the T cells administered to C018 was listed as 'NA'.
4. The % of tetramer+ T cells was very low for several of the patients, including 2 of the PRs, C005 and C020, where it appeared that only ~5% of the infused T cells were tetramer reactive. Given this fact, it appears that the in vitro stimulation was not very efficient in stimulating T cells reactive with these peptides. In addition, it is difficult to attribute the clinical responses to these T cells given the fact that the total number of reactive T cells infused into these patients would have then been extremely low. For patient C005 that number would appear to be ~5% of $\sim 1e8$ or $\sim 5e6$ T cells. Given this data it is difficult to attribute these responses to the infused neoantigen-reactive T cells.

Reviewer #3 (Remarks to the Author):

Thanks and the authors addressed all my concerns.

Reviewer #1 (Remarks to the Author):

I would like to thank the authors for their clear answers, which is very helpful

Reviewer #2 (Remarks to the Author):

The changes made in the manuscript addressed the majority of issues raised by the reviewers, but there are a few points that need to be further addressed.

1. It is not possible with a small study such as this to say that the LD had any effect on responses with the limited number of patients in this study.

Response: Thank the reviewer for the comment. Indeed, we acknowledge that this is a small phase I study. While our findings showed that LD chemo reduce lymphocyte counts, inhibit and delay expansion of neoantigen specific TCR clones and suggest that LD may reduce the efficacy of Neo-T treatment, it is possible that other factors could have contribute to the outcomes. For example, differences in patient characteristics, the tumor microenvironment, the immunogenicity of neoantigens and the amount of neoantigen specific Neo-T cells infused could all have influenced the results. Limited by the size of the current study, we were unable to investigate the effects of all these variables and to generalize our findings to a larger population. Therefore, future investigations with larger sample sizes are necessary to better understand the impact of lymphodepletion on Neo-T treatment responses. We added this to the Discussion section. (lines: 272-281).

2. The information presented in Table S1 needs further clarification. If this is the fraction of CD8+ T cells in the infused products then it should be noted as such and not a ratio. A ratio compares 2 different values and the legend just refers to a single variable, the number of CD8+ T cells.

Response: Thank the reviewer for the comment. We changed the name from "Average ratio of CD8+T in Neo-Ts" to "Average fraction of CD8+T in Neo-Ts" in Table S1.

3. It is not clear why the viability of the T cells administered to C018 was listed as 'NA'.

Response: Thank the reviewer for pointing this out. It was mislabeled as 'NA', we corrected it in this revision.

4. The % of tetramer+ T cells was very low for several of the patients, including 2 of the PRs, C005 and C020, where it appeared that only ~5% of the infused T cells were tetramer reactive. Given this fact, it appears that the in vitro stimulation was not very efficient in stimulating T cells reactive with these peptides. In addition, it is difficult to attribute the clinical responses to these T cells given the fact that the total number of reactive T cells infused into these patients would have then been extremely low. For patient C005 that

number would appear to be ~5% of ~1e8 or ~5e6 T cells. Given this data it is difficult to attribute these responses to the infused neoantigen-reactive T cells.

Response: Thank the reviewer for the comment. Yes, ~5% is not very high, but it is an average number of neoAg specific T cells we often seen in patients when we do in vitro stimulation of T cells. For example, in a previous report, neoantigen-specific T cells were generated by priming T cell from PBMC with one predicted dominant neoepitope, and the fraction of neoantigen-reactive CD8+CD137+T cells in the final T-cell products ranged from 2.14-6.78% in 6 patients. Infusion of these T cell products resulted in clinical responses of 1xCR, 1xPR, 3xSD, 1xND among 6 patients (PMID: 30835255). In another report, the fraction of several neoantigen-specific T cell clones in a TIL product ranged from 0.1% to 11%. Although some are extremely low, they all exhibited strong immunogenicity against tumor cells (PMID: 26389673). Indeed, in our study, the infused number of neoantigen specific T cells were very low in patient C005, but we think these neoantigen specific Neo-T cells can contribute to the clinical responses. Limited by the size of the current study, we were unable to investigate how much these low number of Neo-T cell contribute to the clinical response. Future investigations with larger sample sizes are necessary to provide clear evidence.

Reviewer #3 (Remarks to the Author):

Thanks and the authors addressed all my concerns.